# Long distance particle transport to the central Ionian Sea

Léo Berline[1], Andrea Michelangelo Doglioli[1], Anne Petrenko[1], Stéphanie Barrillon[1], Boris Espinasse[2], Frederic A.C. Le Moigne[1], François Simon-Bot[1], Melilotus Thyssen[1], François Carlotti[1]

[1]Aix Marseille Université, Université de Toulon, CNRS, IRD, Mediterranean Institute of Oceanography (MIO), 13288, Marseille, France
[2]Department of Arctic and Marine Biology, UiT The Arctic University of Norway, Tromsø, Norway

*Correspondence to*: Leo Berline (leo.berline@mio.osupytheas.fr)

**Abstract**

Together with T-S properties, particle abundance *in situ* measurements are useful to discriminate water masses and derive circulation patterns. In the upper layers of the Ionian Sea, the fresher Atlantic Waters (AW) recently crossing the Sicily channel meet the resident and saltier AW which circulated cyclonically in the Eastern basin and modified after evaporation

and eventually cooling.  In May 2017, during the PEACETIME cruise, fluorescence and particle abundance sampled at high resolution revealed unexpected heterogeneity in the central Ionian. Surface salinity measurements, together with altimetry-derived and hull-mounted ADCP currents, describe a zonal pathway of AW entering the Ionian Sea, consistent with the so-called cyclonic mode in the North Ionian Gyre. The ION-Tr transect, located between 19-20°E at approximately 36°N turned out to be at the crossroad of three water masses, mostly coming from the west, north and from an isolated anticyclonic

eddy northeast of ION-Tr. Using Lagrangian numerical simulations, we suggest that the contrast in particle loads along ION-Tr originates from particles transported from these three different water masses. Waters from the west, identified as  AW carried by a strong southwestward jet, were moderate in particle load, probably originating from the Sicily channel. Water mass originating from the north was carrying abundant particles, probably originate from northern Ionian, or further away from the south Adriatic. Waters from the eddy, depleted in particles and Chl-a may originate from south of the Peloponnese,

where the Pelops eddy forms.

The central Ionian Sea hence appears as a mosaic area, where waters of contrasted biological history meet. This contrast is particularly clear in spring, when blooming and non-blooming areas co-occur. Interpreting the complex dynamics of physical-biogeochemical coupling from discrete measurements made at isolated stations at sea is a  challenge. The

combination of multiparametric *in situ* measurements at high resolution with remote sensing and Lagrangian modeling appears as one adequate way to address this challenge.

# 1 Introduction

The study of particle abundance, size structure and dynamics is a key to understand the marine ecosystem, from primary and secondary production to export and remineralization all the way through food-web dynamics (McDonnell et al 2015, Giering et al 2020). In the semi-enclosed Mediterranean Sea, the main particles sources are the biological production in the open ocean and over continental shelves, rivers discharge, atmospheric deposition as well as sediment resuspension. From West to East and North to South, the Mediterranean Sea harbours areas of very distinct productivity, going from eutrophic (e.g. in the Alboran Sea) to ultra oligotrophic (e.g. in the Levantine basin, Moutin et al 2017). This heterogeneity is further enhanced by the complex general and mesoscale circulation that favor exchanges between these contrasted areas, leading to a fine scale spatial distribution of biogeochemical properties and particles (Karageorgis et al 2008, Durrieu de Madron et al, 1992, Rousselet et al 2019).

Located in the center of the Mediterranean Sea, the Ionian Sea is a cross road where three different surface water masses meet: i) recent Atlantic water (AW) flowing from the Sicily channel ii) older AW flowing from the East after its cyclonic circulation in the Levantine basin and iii) fresher (Po influenced) water coming from the Adriatic Sea (Malanotte-Rizzoli et al 1997). Circulation in the North Ionian Sea is strongly variable, as is the path of AW . The most stable branch of the Atlantic Ionian Stream entering the Ionian Sea follows a rather direct pathway to the SouthEast, connecting the Sicily Channel with the Cretan Passage (Mid Ionian Jet, MIJ, Menna et al 2019). South of the MIJ, the circulation pattern is stable and anticylonic, known also as the Syrte Gyre (Pinardi et al, 2015). North of the MIJ, two alternate circulation states have been proposed, the anticyclonic and cyclonic mode of the so called North Ionian Gyre (NIG, Gačić et al 2010). In addition, semi-permanent mesoscale gyres are observed in the western part (Maltese Channel Crest, Medina Gyre, Messina Rise Vortex) as well as in the eastern part (Pelops Gyre), some of them triggered by wind. Although the AW circulation has been documented according to seasons and NIG modes (Malanotte-Rizzoli et al 1997, Gačić et al 2011, Menna et al 2019), the fine-scale pathways of AW crossing the Ionian Sea have only seldom been sampled. The mode of the NIG has a strong influence on the dispersal of water masses and properties in the Ionian basin, also impacting its productivity (Lavigne et al 2018).

The Ionian Sea is generally considered oligotrophic (Boldrin et al 2002), with a north-south gradient of Chl-a (d'Ortenzio et Ribera d'Alcala 2009). However, it is not homogeneous, as distinct phytoplankton communities associated to the main water masses have been described (Casotti et al 2003). Three main phytoplankton communities were associated with water coming from Adriatic in the north west, water from the Eastern Mediterranean in the north east and AW from the Sicily channel to the south. Zooplankton community was also contrasted between the northwestern and eastern Ionian Sea (Mazzocchi et al 2003).

Particle distribution patterns in the Ionian Sea have been seldomly described. The particle concentration in the water column can be measured with several instruments. Discrete sampling is carried out using bottles or pumps followed by filtration,

allowing bulk mass measurements (McDonnell et al 2015). Continuous sampling can be carried out using optical measurements that give either bulk measurements (transmissometer, backscatter (Briggs et al 2013)) or size-resolved measurements: Laser In Situ Scattering and Transmissometry- LISST, (Karageorgis et al 2012), Underwater Vision Profiler – UVP (Picheral et al 2010), Laser Optical Particle Counter – LOPC, (Herman, 2004) for the most common instruments, see Giering et al (2020) for a comprehensive review. In the Ionian Sea, few data are available on the horizontal distribution of particles. These data have a coarse horizontal resolution and are mostly from transmissometry, water filtration (Rabitti et al 1994, Boldrin et al 2002, Karageorgis et al 2008, 2012) while few data are from optical devices (Karageorgis et al 2012, Ramondenc et al 2016). Here we used the LOPC, providing size resolved abundances over the size range [100-2500] µm. This instrument was mounted on a free-fall MVP fish, allowing high resolution horizontal and vertical sampling of particles and water mass properties along the ship track.

The objective of this paper is to document particle distribution in the central Ionian Sea, based on a high-resolution multi-parametric transect. In particular we analyse particle distribution with regard to water mass properties and transport, to propose scenarios of histories of these waters masses and discuss implications for the functioning of the Ionian Sea ecosystem.

## 2 Material and Methods

The PEACETIME cruise (http://peacetime-project.org/, Guieu et al 2020) was conducted during late spring conditions from May 10th to June 11th, 2017 on board the R/V *Pourquoi Pas ?*. The overall aim of PEACETIME was to investigate the role of atmospheric dust deposition on biogeochemical fluxes and pelagic ecosystem functioning. Along the 4300 km transect covering western and eastern Mediterranean basin, 10 short stations (with an average duration of 8 hours) and 3 long stations (duration of 4-5 days) were carried out. The ship entered the Ionian Sea on May 22nd and the last station (ST8) in the region was occupied on May 30th as detailed in table 1. An additional station, the SAV station at a distance from the main track was was chosen for a float deployment.

At stations, Rosette casts were carried out (0-500 and 0-bottom). The system was equipped with pressure, temperature conductivity (Sea-Bird SBE9), dissolved oxygen concentration (SBE43), photosynthetically active radiation (LI-COR Biospherical PAR), light transmission at 650 nm (Sea-Bird C-Star, Karageorgis et al 2008) and chlorophyll a fluorescence (Chelsea Acquatracka). Fluorescence was calibrated to report Chl-a in mg.m$^{-3}$. This instrumental package included 24 12-L Niskin bottles. Water samples collected with the classical Rosette was used to quantify oxygen, nutrients, Dissolved Organic Carbon, Particulate Organic Carbon and Nitrogen and pigments using HPLC (see Guieu et al 2020 and Marañón et al 2020 for detail on HPLC).

## 2.1 High resolution vertical sampling

Between stations, measurments with a Moving Vessel Profiler (MVP) were carried out. two transects , ION-Tr and a very short ST8-Tr were carried out (figure 1). MVP200 ODIM Brooke Ocean performed vertical profiles from surface to ~300 m, at a spatial resolution of ~1 nm, with the nearly vertical free falls of the Multi Sensor Free Fall Fish type I equipped with a AML microCTD, a WetLabs ECO fluorimeter, and a ODIM Brooke Ocean Laser Optical Plankton Counter (LOPC). The T and S of the microCTD were calibrated with the rosette CTD. The LOPC recorded particle size and abundance distributions: the instrument records the cross-sectional area of each particle passing through its laser beam (Herman et al., 2004; Herman & Harvey, 2006) for a size range between 100 µ m and a few centimeters. If a particle is recorded by at least 3 diode elements, it will be recorded as a multi-element plankton (MEP), in contrast to single element plankton (SEP). LOPC also provides information about the MEPs allowing to compute an attenuance index (AI), which is essentially a normalized opacity index (an AI of 1 means a fully opaque object) although the MEP shape also has an impact (Espinasse et al 2018). The living fraction of particles can be estimated, using a calibration based on comparison with plankton net tows (Espinasse et al., 2018), as this fraction is linked to the percent contribution of MEP with respect to the total particle count (%MEP). The sampling volume was estimated using the LOPC sampling tunnel surface multiplied by the depth increment estimated with the pressure sensor. Given the fish free-fall velocity of ~4 m.s$^{-1}$ and the LOPC acquisition frequency (2 Hz), we binned the LOPC data in 5-m vertical bins. As the surface docking depth of the MVP fish was not constant, the first bin (0–7.5 m) was discarded from the analysis. The LOPC abundances were redistributed into log-scale size bins for easier analysis. Total particle abundance (e.g., abundance over the full LOPC size range) and parameters of the size distribution were computed (slope and intercept of the Normalized Biomass Size Spectra (NBSS), e.g. Espinasse et al., 2018). We thus have access to vertical profiles of total abundance of particles (particles per cubic meter) at almost each kilometer during the MVP transects.

## 2.2 Underway measurements

A ThermoSalinoGraph (TSG), equipped with a SeaBird SBE21, was connected to a continuous surface water flow-through system that pumped seawater at 2-m depth. Output data were sea surface temperature and Salinity every minute all along the cruise route. Horizontal velocity was measured with a hull mounted ADCP.: RDI Ocean Surveyor 150kHz. Data treatment was carried out using the CASCADE software version 7.1 (Le Bot et al 2011) and LATEXtools toolbox (Petrenko et al., 2017, Doglioli et al., 2013, https://people.mio.osupytheas.fr/~doglioli/latextools.htm).

## 2.3 Automated cytometry

Surface phytoplankton community was analysed using an automated flow cytometer (Cytosense benchtop flow cytometer from CytoBuoy b.v) installed on a dedicated continuous sampling system set up to pump surface water (at 5m depth). The Cytosense automated flow cytometer was equipped with a 120 mW, 488 nm laser beam. The volume analyzed was

controlled by a calibrated peristaltic pump and each particle passed in front of the laser beam at a speed of 2 m.s$^{-1}$. The
particle resolved size range varied from <1 µm up to 800 µm in width and several hundreds of µm in length for chain
forming cells. The trigger to record a signal was based on pigment fluorescence. Stability of the optical unit was controlled
before and after the cruise using fluorescing 2 µm Polyscience beads, and a set of silica beads of 1.0 , 2.0 ,3.0, 5.0, 7.0 µm
diameter, respectively. Cell size were estimated from the forward scatter (FWS signature) of the silica beads , and enabled to
separate pico, nano and microphytoplankton size classes. The instrument allowed for the quantification of pico-, and
nanophytoplankton populations, up to microphytoplankton when abundant enough in the 5 mL analyzed. Further, seven
groups were resolved. One group of picoeukaryotes (mean size 2.81±0.19 µm), three groups of nanoeukaryotes
(Nanoeukaryotes1, 2 and 3, mean sizes of 4.76±0.13, 7.16± 0.35 and  10.64 ± 2.18 µm respectively), Synechococcus like
(1.04± 0.12 µm), Chryptophytes like (6.00 ±0.93 µm) and a coccolitophore like group (5.37± 0.14 µm). Each phytoplankton
group was manually clustered using the Cytoclus® (http://www.cytobuoy.com/products/software/) dedicated software.

**2.4 Satellite data**

Several remote sensing datasets were exploited using the SPASSO (Software Package for an Adaptive Satellite-based
Sampling  for  Ocean  campaigns, https://spasso.mio.osupytheas.fr/; last access: 2 Oct. 2020). Altimetry data was obtained
from the AVISO Mediterranean regional product. Delayed-time L3 maps of SST Sat  (Mediterranean Sea – Ultra High
Resolution L3 SST Reprocessed; Nardelli et al., 2013; Pisano et al., 2016) and Chl Sat a (Mediterranean Sea Reprocessed
Surface Chlorophyll Concentration from Multi Satellite observations) were retrieved for the period of the cruise, on regular
grids of 0.01 × 0.01 ◦ resolution for SST Sat and of 1  ×  1 km resolution for Chl Sat a from CMEMS - Copernicus Marine
Environment Monitoring Service  (http://marine.copernicus.eu/). SPASSO provided several Lagrangian diagnostics of the
altimetry-derived current field, such as the Finite scale Lyapunov Exponent (FSLE) computed using the algorithm of
d'Ovidio et al. (2004), in order to help in positioning the sampling transect and stations taking into account the ocean
dynamics. This approach was successfully adopted during previous cruises such as LATEX (Petrenko et al, 2017), KEOPS2
(d'Ovidio et al., 2015), OUTPACE (Moutin et al, 2017).

**2.5 Lagrangian numerical simulations**

In order to estimate the origins of the sampled surface water along transect ION-Tr,  the ARIANE package (Blanke and
Raynaud, 1997) was used to predict surface trajectories of water particles, using surface geostrophic velocities derived from
altimetry and winds from the WRF model (Barrillon et al 2019). The daily gridded surface geostrophic velocity fields from
Copernicus Monitoring Environment Marine Service (CMEMS) were used, with a resolution of 1/8°, using the near real time
data, validated by an accurate comparison with both the hull-mounted ADCP-data and the Surface Velocity Program (SVP)
drifters deployed during the cruise (Barrillon et al 2019).  Given the daily time series of the surface velocity field, the
ARIANE package computes numerical particle trajectories through a backward or forward integration time. The so-called
"qualitative" mode is chosen: each particle trajectory can be followed step by step through the integration time. Only surface

trajectories were considered. The simulated particle positions were initialized in a polygon around the ship's track on May 29[th] (longitude from 18° to 20.1° E, latitude from 35.4° to 35.9° N with an extension of 0.2° north and south of the track. ). This polygon surrounds the actual track, to take into account a potential offset between *in situ* velocity and satellite derived geostrophic velocity. 10000 particles (sufficient for a comprehensive sampling of the polygon area) randomly seeded inside the launch area were advected backward for one month. This duration was chosen as a trade-off between particle source identification,  error accumulation along the trajectories and residence time in the Ionian Sea (39 days according to Celentano et al 2020). The particle trajectories and final positions were analyzed to determine the main source areas.

The cruise data are available on the LEFE-CYBER database (http://www.obs-vlfr.fr/proof/php/PEACETIME/peacetime.php).

Once the special issue is completed, the PEACETIME biogeochemical dataset will be available in SEANOE at https://www.seanoe.org/data/00645/75747/

## 3 Results

### 3.1 Oceanographic context

Considering the surface circulation over the northern Ionian Sea (figure 1), averages of satellite Chl-a and surface temperature fo the region 16° to 22°E, 34° to 40°N were computed for the period before our sampling. Time series of satellite Chl-a showed that surface phytoplankton concentration decreased since mid-March (figure 2). Maximum Chl-a concentration was 0.24 mg.m$^{-3}$ in January. Since the end of April, the Ionian Sea was steadily warming while Chl-a concentration remained low (0.09 mg.m$^{-3}$) indicating the absence of significant surface production. On average over April-May, maximum Chl-a concentrations were observed in the area South of the Strait of Messina and in the Gulf of Taranto (not shown).

At the end of May in the Ionian basin, satellite imagery (figures 3A and 3B) shows  slight  meridional gradients of SST and Chla, with warmer and low Chl water to the south and colder and water with higher Chl to the north. In addition, a strong mesoscale variability was observed  with cold and Chl-rich filaments south of the Strait of Messina and south of Sicily, extending toward the center of the basin. Several cyclonic (located at 38°N 17.5°E and 34.5°N 21.3°E) and anti-cyclonic (located at 35.7°N 19°E, 34.7°N 17.5°E) structures were visible in the center of the basin.

The underway salinity showed a general increase toward the east and the north. On the northern part of the route (SAV-ST7-ION), there was a transition from medium salinity to the north to low (<38.2) then high (>38.8) salinity at the easternmost end of the route close to the long duration station ION. On the southern part of the route (ION to ST8), there was an alternation of low and high salinity waters, sampled during the ION-Tr transect.  West of 18°E, salinity was stable and lower

than 38.2 up to the Sicily channel. Therefore SAV and ST7 were in medium salinity waters, ION was in high and ST8 was in low salinity waters, while ION-Tr crossed waters with variable salinities.

The two phytoplankton groups showing c abundant enough for the counts to be accurate are shown (figures 3C and 3D). In the Ionian Sea, nanoeukaryotes1 and coccolithophore like cells showed a distribution consistent with salinity, with higher values in saltier water except southeast of Sicily. The patterns observed were mostly consistent with satellite Chl-a, with high abundances southeast of Sicily and south of the Strait of Messina. However, along ION-Tr, abundance was variable while
satellite Chl-a was fairly constant.

Altimetry-derived surface geostrophic currents showed a general circulation with an anticyclonic flow in the southwestern part of the basin (figure 4A). South of Sicily, water entered the Ionian Sea around 36°N along a southeastward path, progressed eastward along 35.5°N, then veered southwestward close to 19.5°E. East of 18°E, the currents show a pattern
consistent with the intense (~0,5 m.s$^{-1}$) currents measured by the hull-mounted ADCP along the ship route, depicting an anti-cyclonic meander, more intense on its eastern flank. There is a small spatial offset between altimetry and ADCP (~0.4° longitude, figure 4B), and current magnitude estimated from AVISO was about half the value of ADCP derived current magnitude. The southwestward branch of this meander was located approximately in the center of the ION-Tr transect, while the ION station was located on the eastern border of the meander. Altimetry-derived FSLE field (figure 3), as well as the
time series of ADT (annex A) showed that this anticyclonic meander east of 18°E was stable during May. The southwestward jet sampled during ION-Tr coincided with a slightly cooler filament than surrounding waters, as shown on figures 3A and 3B.

### 3.2 Distribution of water masses and biological properties along transect ION-Tr

Based on the salinity depth distribution (figure 5), ION-Tr can be divided into three parts. From West to East, a western part
(noted W) with a fresher surface layer extending 0-30 m covering a deeper layer with salinity >39, , a center part (noted C) with a fresher  vein (<38.7) extending from surface down to 70 m deep, and an eastern part (noted E) with quasi-homogeneous salinity on the vertical, slightly lower than in W (39). Hereafter we define three pseudo-stations along ION-Tr, named ION-W, ION-C, ION-E. Note that the ION long duration station was located very close to the eastern part of ION-Tr (figure 1), thus ION-E and ION can be considered as similar.

This salinity  coincided with current pattern from the ADCP , with ION-W and ION-E being in quiescent water (current<0.2 m.s$^{-1}$), while ION-C correspond to a strong (>0.3 m.s$^{-1}$) southwestward jet extending from the surface to 300 m depth. The total abundance of particles, as well as Chl-a fluorescence clearly contrasted between ION-W and ION-E (Figure 5). In ION-W, particle abundance was low (<5 10$^4$ #.m$^{-3}$), essentially concentrated within the 20-30 m layer. Fluorescence was rather homogeneous with a broad maxima around 110 m. In ION-E part, abundance was much higher (>2 10$^4$ #.m$^{-3}$) with a deep

maximum at 100 m depth (values around $1 \cdot 10^5$ #.m$^{-3}$) close to the depth of the Deep Chlorophyll Maximum (DCM) and also high particle abundance between 20 and 90 m (~ $3 \cdot 10^4$ #.m$^{-3}$). ION-C was distinct from ION-E and ION-W in terms of vertical distribution and abundance of particle as well as fluorescence, with the low salinity vein coinciding with higher particle abundances and the maximum fluorescence being located ca 70 m, at the interface between low salinity and underlying waters.

The T-S distribution of ION-Tr (figure 6) mirrored the distributions of stations ST8 and ION. Water at ION-C is similar to ST8 for potential densities greater than 1028 kg.m$^{-3}$ (ie deeper than 25m). ION-E is similar to ION with more scatter. ION-W does not match the other stations. Northern stations ST7 and SAV had similar water masses properties, distinct from ST8, ION and the ION-Tr profiles.

At the surface, phytoplankton groups also showed contrasting abundances along ION-Tr (figures 3C and 3D). Horizontal sampling resolution did not allow to distinguish the ION-C region.  But nanoeukaryotes1 increased from low values (as mean ± SD, 487±52 cells.cm$^{-3}$, n=27, 18-19.5°E) to higher values (659 ± 50 cells.cm$^{-3}$, n=117, >19°5E) along ION-Tr (from west to east). Similarly, coccolithophore like group also increased (176 ± 18 cells.cm$^{-3}$ against 192.47 ± 13 cells.cm$^{-3}$). Conversely, picoeukaryotes abundances decreased eastward (653 ±94 cells.cm$^{-3}$ vs 462.8±76 cells.cm$^{-3}$, not shown). The other groups did not show large changes in abundances.

### 3.3 Comparison of vertical distribution across stations

CTD and bottle measurements performed at the SAV, ST7, ST8 stations together with the one from the ION transect are shown in figure 7. These vertical profiles allow a finer analysis of the contrast between stations and among parts of ION-Tr. Consistent with figure 6, Salinity increased from ST8 to ION-E (figure 7, panels A, F). Excluding the upper  25 m of the water column corresponding to the mixed layer (ML), couples of stations were similar: ION-C and ST8, ION-E and ION-W, ST7 and SAV. A narrow (<10 m thick), lower salinity layer was present at 30 m on every profile except ION-E. The surface pycnocline was located between 10 to 25 m depth, followed by a steep thermocline down to 60-80 m. Surface nitrate (figure 7, panel C) was depleted everywhere. Nitracline depths increased from 60 m at SAV up to 90 m at ION, together with nitrate concentrations below the euphotic zone.

A DCM was always present (figure 7, panels D, H), but its intensity was highest at ST8 and lowest at ION-W and ION-C, with fluorescence expanded over a large depth range. The depth of DCM corresponded to the top of nitracline and to the bottom of the euphotic zone (~90 m at ION).

Profiles of particle abundance were highly variable (figure 7, panel E, I). The integrated particle abundance was highest at SAV and lowest at ION-W. Some profiles had several maxima. At ST8 and ION-E, the main peak was just above the DCM depth, and a secondary peak was present above DCM at ~20 m depth, also present at ION-W. The SAV profile was different with a large peak at 50 m, above DCM, and a surface peak at 10 m.

The NBSS slopes (table 2) were similar for stations SAV, ION-W and ION-E (0.85), higher for ST8 (0.93) and intermediate for ION-C (0.89) indicating more abundant smaller particles at ST8. On average, using the calibration given in Espinasse et al (2018), %MEP indicated that at ION-C and ION-W, zooplankton constituted ~ 60% of particles, while at ION-E, ST8 and SAV particles were mainly detritus (~75-90%) ie non-living particles such as aggregates produced after intense primary production. AI was similar in different depth layers but higher at ST8 and ION-E (0.11-0.10), while MEP was always more abundant in the 50-150 m layer, where the particle maxima were observed.

### 3.4 Lagrangian origin of particles

The focus here is on the particles produced before sampling and transported to the region of sampling. Figure 8 show the full trajectories of particles originating along the ION transect, splitted in four panels according to the main path followed by particles and therefore their origin. Each color identifies a distinct path. Panel E show the particle positions one month before our sampling, colored according to the origin. Note that as the altimetry-derived velocity was shifted westward by 0.4° in longitude west with respect to *in situ* velocity from ADCP (figure 4B), the initial positions of numerical particles were extended westward to 18°E. These simulations showed that the water parcels observed along ION-Tr originate from four regions, so called source regions (South origin So, West origin Wo, North origin No, Eddy origin Eo) with Wo, No and Eo being dominant (Figure 4 F). Along ION-Tr, the contribution of each region showed a rather clear pattern, with some overlap in the center (figure 8F). ION-W was in waters mainly originating from Eo. ION-C was mainly originating from Wo but also from Eo and to a lesser extent from So. ION-E was only originating from No. Finally, west of 18.6°E, beyond the limit of the actual transect, So was the main contributor.

## 4 Discussion

### 4.1 Surface circulation and distribution of water masses in the Ionian Sea

Surface circulation, together with the along track surface salinity distribution, describe one pathway of AW penetrating into the Ionian Sea (figures 3 and 4). Here, this pathway is approximately zonal along latitude 36°N from the Sicily Strait up to 19°E, then veering southwestward. This pathway is supported by the similarity of water masses properties sampled at ST8 and ION-C (figures 6 and 7) and illustrated by backtracked trajectories (figure 8, Western origin), indicating a *ca* one month travel time between 15°E and 19.5°E. It is also supported by the absence of transport barriers east of ST8 shown by FSLE (figure 3). North of 36°N, the similarity of ST7 and SAV water masses properties (figure 6) also supports a single origin of water masses in the Northern Ionian.

This distribution of AW in the Ionian Sea is consistent with a cyclonic mode of the so called North Ionian Gyre (NIG, Gačić et al 2010), that characterizes the circulation in the Ionian Sea north of *ca* 35.5°N. In the cyclonic mode of the NIG, AW

follows a rather direct East-SouthEast path, while saline water from the Levantine basin enters the Northern Ionian Sea from the East, and current flows southwestward along Calabria (Menna et al 2019). As observed by altimetry, the NIG was cyclonic since 2011 (Menna et al 2019), and reversed to anticyclonic in the second half of 2017 (von Schuckmann et al., 2019), after our sampling.

It is interesting to note that the southwestward jet sampled along ION-Tr is a persistent feature of the Ionian Sea circulation, generally located at ca 20°E, 36°N and directed southward (see figure 2 in Menna et al 2019, Celentano et al 2020, Berline et al 2014). It is present during cyclonic and anticyclonic modes of the NIG. It pertains to the northern branch of the Mid Ionian Jet (MIJ).

Cyclonic circulation of the NIG implies a downwelling of the nutricline at the northern border of the NIG, and an upwelling of nutricline in the central NIG (Civitarese et al 2010, Lavigne et al, 2018). Our observations of shallow nutricline depth (isoline 1 µM at ~55 m for SAV and ~95 m for ST8, both located in central NIG) are consistent with the nutriclines shown by Lavigne et al, (2018) during cyclonic mode. Cyclonic mode also favors higher Chl-a in the center NIG, and a late winter bloom (Lavigne et al, 2018). However the Chl-a evolution during 2017 only shows a weak bloom in the Northern Ionian Sea (figure 2). This may result from a weak winter buoyancy loss, independent of the NIG circulation, as pointed out by Lavigne et al (2018). The increase of Chl-a occurred in the last week of April 2017, then Chl-a decreased reaching its minimum value around the end of May, during our sampling.

### 4.2 Origin of water masses sampled along ION-Tr

In the center of ION-Tr, a strong current is associated to a low salinity vein (down to 38.4, figure 6), that dynamically separates the two parts of the transect. This separation is stable as supported by the FSLE ridge (figure 3). While the density stratification is similar along the transect, particle and Chla distributions show a striking difference from E to W. At ION-E, there is 2.8 times higher particle abundance, and 2.4 times higher biovolume than at ION-W (table 2), essentially due to particles with and equivalent spherical diameter (ESD) smaller than 500 µm. At ION-E particles were distributed over the pycnocline from 1028 to 1028.7 kg.m$^{-3}$ (potential density, figure 5d), with a maximum associated to a narrow DCM at 100 m, while at ION-W particles are only present in the density layer 1028-1028.2 kg.m$^{-3}$, and the DCM is broad and less marked, with no significant particle peak.

The distinct Chl-a and particle distributions along ION-Tr suggest that waters sampled at ION-E and ION-W have distinct histories of biological production. One hypothesis is that these water masses originate from distinct locations. To explore this hypothesis, backtracking of particle trajectories was carried out and led to three main origins of water sampled along ION-Tr (Wo, No and Eo, as So is only marginally contributing to the transect), influencing different parts of the transect (figure 8). When two origins overlap, we only consider the dominant origin. Water found at ION-C would be transported from the West

(Wo), ie the Sicily channel, while water found at ION-E (ION-W) would be transported mostly from the Northern Ionian (No), and from an eddy located North (Eo) respectively. Referring back to the general circulation in the Ionian Sea, the ION-Tr appears to be at the crossroad between waters from the North Ionian Sea, waters from the MIJ coming from the West and waters trapped into an anticyclonic eddy.

## 4.3 Biological history of water masses sampled along ION-Tr

LOPC detects particles of ESD >100µm (median ESD was 130-145 µm along ION-Tr). Typically, such large aggregates form as a result of intense primary production (Briggs et al., 2020; Martin et al., 2011). In addition, the presence of transparent exopolymer particles (so called TEP) represents an important upper ocean process by which dissolved organic matter is converted into particles. Besides, these compounds act as glue enhancing small particles aggregation and leading the formation of marine snow aggregates (Engel et al., 2004) of the size we observe here. Once produced, aggregates can
fragment or coagulate, be grazed, be degraded by bacteria attached to them and sink out of the surface layer (Le Moigne 2019, Stemmann et al 2004). Therefore the particle abundance in the euphotic layer generally peaks shortly after enhanced primary production, then decreases in a time span depending on several factors.

Although the age of particles cannot be determined from their size distribution, we can make some hypothesis. At the time of
sampling, the whole Ionian Sea was stratified with shallow mixed layers  (~25 m) and nitrate was fully depleted in the euphotic zone except at SAV, indicating stable, post-bloom conditions. As surface Chl-a continuously decreased since the end of April (figure 2), this suggests that the last surface primary production event around ION station occurred about a month before our sampling. This is longer than the time lag observed between primary production and subsequent export in large mesocosms experiments (Stange et al., 2017). Therefore, freshly formed particles are not likely.

The sinking speed of particles depends on their mineral composition (Le Moigne 2013), shape (Laurenceau et al., 2019) and on the phytoplankton community (Riley et al 2012, Guidi et al 2009). Consistent with our cell counts, the pigment composition available at ST7, ST8 and ION show a community dominated by nanophytoplankton (>60% of total Chl-a) in the upper 80 m, while diatoms were abundant at the DCM depth (41-51% of total Chl-a, see Marañón et al 2020). The
dominance of nanophytoplankton and coccolithophores was also found by Boldrin et al (2002) in the Northern Ionian Sea (38.5°N). Thus, in the upper 80 m small cells (typically <5 µm in diameter) do not favor the production of large aggregates and direct export (Richardson and Jackson 2007).

Further, aggregates are slowed down by density gradients, because of the equilibration time of their interstitial water
(MacIntyre et al 1995, Prairie et al 2015). Therefore pronounced stratification can locally increase their abundance, as reported in numerous settings (Espinasse et al 2014, Marcolin et al 2013, Ohman et al 2012). This explains that all profiles

have a particle peak, just below the mixed layer depth (figure 7). However, this mechanism does not explain deeper peaks observed at 60 m and 90 m.

Finally, zooplankton can also contribute to the particle production through fecal pellet and transformation (sloppy feeding), unfortunately we are lacking zooplankton abundance data for ION-C and ION-W.

We now review the potential processes that led to the observed particles and Chl-a distributions along ION-Tr, from west to east.

**4.3.1 Western part**

At ION-W, below the weak particle peak (figure 7, $2 \cdot 10^4$ #.m$^{-3}$) associated with the density gradient at the bottom of the mixed layer, abundances steeply decrease and reach minimum values at 80 m ($1 \cdot 10^4$ #.m$^{-3}$). Such a low particle abundance in the top 100 m was not observed anywhere else across all PEACETIME LOPC profiles (~800 casts, Guieu et al, 2020). The Chl-a concentration is also lower and very patchy compared to ION-E. As nitracline depth was shown to be highly correlated

with the depth of the 1028.9 kg.m$^{-3}$ isopycnal in the Ionian Sea (Lavigne et al 2018), the steep deepening of this isopycnal from ION-E to ION-W suggests a much deeper nutricline at ION-W than at ION-E (figure 5). Together, these observations emphasize the very oligotrophic character of station ION-W. Considering the backward trajectories (figure 8), this low abundance may result from two origins (i) waters trapped into an anticyclonic eddy (Eo), or (ii) waters from the southern Ionian Sea (So), which is more oligotrophic and where phytoplankton concentration peak earlier in the year (D'Ortenzio and

Ribera d'Alcala 2009). For hypothesis (i), analysis of the evolution of altimetry (annex A) revealed that the eddy stayed coherent during the first six months of 2017 while slowly moving from the eastern Ionian. This eddy may have trapped waters from the more oligotrophic eastern Ionian Sea (Casotti et al 2003), and the vertical velocities in the center of the anticyclonic eddy may have accelerated the export of particles, as observed in the eastern Mediterranean (Waite et al 2016). For hypothesis (ii), it was not supported by particles profiles obtained with Underwater Vision Profiler (UVP) in the

southern Ionian in July 2008 (Picheral et al 2010) (ecotaxa.obs-vlfr.fr, Kiko et al 2020), as they did not show such low abundances.

**4.3.2 Center part**

At ION-C, the backtracked trajectories indicate two overlapping origins (Eo and Wo mainly). However the low salinity vein extending down to 70 m is a clear footprint of AW observed at ST8. Given the distribution of surface salinity from the

375 thermosalinograph (figure 3A and 3B) this gives more support to a Wo. ST8 and ION-C also share higher NBSS slopes (0.93 and 0.89). As with ION-E, significant particles abundance above the DCM suggest that particles were produced upstream, near ST8 or further west, possibly in the enriched area south of Sicily, then advected to central Ionian Sea.

### 4.3.3 Eastern part (ION station)

At ION-E DCM, microphytoplankton was dominant (Marañón et al 2020) and associated to a particle peak in the DCM. This
high ratio of particle to phytoplankton was also noted using other optical instruments (Barbieux et al 2021). The DCM was
associated with the bottom of the euphotic zone (~90 m at ION), but not to a density gradient. Therefore the particle peak
associated with the DCM probably resulted from aggregates produced locally, from diatoms decay and fecal pellet
production by zooplankton. The    association of DCM and  particle peak was not observed in similar settings (see station A
in Espinasse et al 2018), which suggests that the plankton community at the ION-E DCM enhances aggregate production,
perhaps through mucous production, as mucous was reported in sediment traps in the Northern Ionian (Boldrin et al 2002).

Above the DCM, significant particle abundance (2-3 $10^4$ #.m$^{-3}$) was observed together with phytoplankton dominated by
nanophytoplankton (Marañón et al 2020). This significant particle abundance above the DCM is also observed at SAV and
ST8, with much higher peak concentration at SAV than at ST8 (1.5 $10^5$ #.m$^{-3}$ and 3-4 $10^4$ #.m$^{-3}$ respectively). When particles
are suspended or sink slowly compared to the horizontal velocity, particle abundance act as tracer for horizontal advection
(e.g. Karageorgis et al 2012, Chronis et al 2000). We argue that at stations SAV, ST8 as well as at ION-E, the abundance of
particles above the DCM is a remainder of past surface production events that has not been exported out of the surface layer
yet. In the Northern Ionian, a POM maximum above the DCM (0-50 m) was also reported by Rabitti et al (1994), and
attributed to previous (relict) bloom. As above DCM waters are depleted in nutrients, above-DCM particles at ION-E could
be transported from an area where phytoplankton production was more intense and shallower, such as SAV, according to the
backward trajectories and the vertical profiles. This continuity is supported by the similar slope of the particle size spectra at
SAV and ION-E (0.86 and 0.85, table 2). It is also supported by the higher surface abundance of nanoeukaryotes and
coccolithophores at SAV, along Calabria and at ION-E (figures 3C, 3D). During transport from SAV to ION-E, particle
abundance above the DCM may have decreased because of the processes mentioned earlier.

In the upper 0-90 m layer, nutrients were depleted and nanophytoplankton was dominant. Nanoeukaryotes abundances were
stable during the five days of occupation of the ION station (data not shown). This implies an active nutrient recycling,
probably linked to the particle associated microbial community, also supported by the high bacterial production in the upper
100 m of the water column reported by Marañón et al (2020) at ION.

ION-E particle characteristics are closest to the 'continental shelf' habitat type as defined by Espinasse et al (2014), in the
Gulf of Lion in early May with half the integrated particle abundance (18.2 $10^3$ #.m$^{-2}$ vs 38 $10^3$ #.m$^{-2}$), half the average AI
(0.1 vs 0.2), higher %MEP  (6.1 vs 1.8), and similar but steeper NBSS slope (0.85 vs 0.79).  Zooplankton abundance was
slightly higher (300 $10^3$ ind.m$^{-2}$ vs 206 $10^3$ ind.m$^{-2}$, see Feliu et al 2020, but this value may be somewhat higher due to the

combination of two mesh sizes (100 and 200 µm) instead of one (120 µm in Espinasse et al 2018). Thus ION-E was
characterized by porous aggregates, with a rather high contribution of large aggregates (%MEP).

Taking into account Chl-a, nanophytoplankton abundances, significant particle load above the DCM and high bacterial production, ION station was not characteristically oligotrophic, under the influence of particles possibly transported from North Ionian.

**5 Conclusions**

High resolution sampling of water masses, including fluorescence and particle contents revealed unexpected heterogeneity in the central Ionian Sea. Surface salinity measurements, together with altimetry and ADCP-derived currents, describe a zonal pathway of AW entering into the Ionian Sea, consistent with the cyclonic mode of the North Ionian Gyre. The ION-Tr appears precisely located at the crossroad of three water masses, coming from the West, North and from an isolated anticyclonic eddy. Water mass originating from the north carried abundant particles, probably originating from the North
Ionian Sea, or further from the Adriatic. Waters from the eddy were depleted in particles, and probably nutrients, and may originate from the Pelops eddy area, south of Peloponnese. Waters from the West, carried by a strong jet, were intermediate in particle load, probably originating from the Sicily channel, carried by AW.

A particle properties signature for different water masses can be drawn (figure 7 and table 2). Comparing the three water masses observed at ST7-SAV, ST8 and ION, showed three distinct particle signatures in terms of vertical profile, abundance,
425  %MEP and AI, especially for the upprt (5-50 m depth) layer which has the most contrasted particle loads. Comparing water masses salinity and temperature along ION-Tr, only two distinct water masses emerge with (ION-C different from the other two regions. In this case, particle signatures allow to distinguish ION-E and ION-W, with distinct vertical profiles, %MEP and AI. Thus particle properties are complementary to water masses properties as they provide information on their biological history.

Even away from the coasts, the central Ionian Sea appears as a mosaic area, where waters of contrasted biological history meet. ION station is influenced by particles potentially transported from the North Ionian Sea. This contrast is probably amplified in spring, when blooming and non-blooming areas co-occur. Long distance particle transport appears as a significant contribution to particulate matter load, together with atmospheric input.

The small scale heterogeneity of particle abundance revealed here emphasizes the spatial decoupling between particle production and particle distribution. Such decoupling added to the time lag between production and export of particles (Stange et al 2017) may have large impact on assessing the efficiency of carbon export from the surface ocean (Henson et al 2011). This also implies that neutrally buoyant particles can sustain production away from their source. Interpreting the

complex dynamics of physical-biogeochemical coupling from discrete measurements made at isolated stations at sea is a
440 challenge. The combination of multiparametric in-situ measurements at high resolution with remote sensing and Lagrangian modeling appears as one proper way to address this challenge.

**Author contribution**

LB : Conceptualization, Writing – original draft preparation, Formal analysis, FSB : Software, Formal analysis, AD, AP, SB, BE, FLM, FC, MT: Formal analysis, Writing – review & editing.

**Acknowledgments**

This study is a contribution to the PEACETIME project (http://peacetime-project.org; last access xx/xx/2021), a joint initiative of the MERMEX and ChArMEx components supported by CNRS-INSU, IFREMER, CEA and Météo-France as part of the decadal programme MISTRALS coordinated by INSU. PEACETIME was endorsed as a process study by GEOTRACES and is also a contribution to IMBER and SOLAS international programs.

(PEACETIME cruise https://doi.org/10.17600/17000300). Chief scientists and crew are thanked. Jean-Luc Fuda is thanked for his help on CTD calibration.We thank Dominique Lefèvre, Olivier Grosso and Gérald Grégori (MIO) for the set up and handling (DL) of the automated and continuous sea water surface sampling system. We also thank DL and Thibault Wagener for regular checks on the Cytosense flow cytometer during the cruise. GG contributed to the installation and uninstallation of the cytometer.

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

| Station | Arrival date – local time | Departure date– local time | Depth (m) | Lat N | Long E |
|---------|---------------------------|----------------------------|-----------|-------|--------|
| SAV | 23/05/2017 11:30 | 23/05/2017 14:17 | 2945 | 37°50.4 | 17°36.4 |
| ST7 | 23/05/2017 21:10 | 24/05/2017 07:15 | 3627 | 36°39.5 | 18°09.3 |
| ION | 24/05/2017 18:02 | 28/05/2017 20:54 | 3054 | 35°29.4 | 19°46.57 |
| ION-Tr | 28/05/2017 19:41 | 29/05/2017 14:51 | 2980-3520 | 35°21.0 35°38.81 | 20°03.3 18°52.2 |
| ST8-Tr | 30/05/2017 00:43 | 30/05/2017 01:33 | 3200-3314 | 36°10.3 36°12.6 | 16°47.5 16°37.5 |
| ST8 | 30/05/2017 03:53 | 30/05/2017 09:41 | 3314 | 36°12.6 | 16°37.5 |

**Table 1. Sampling in the Ionian Sea (see figure 1). Note that ST8 and ST8-Tr are considered in the study as one single station.**

| | ION-W (n=13) | ION-C (n=10) | ION-E (n=15) | ST8 (n=8) | SAV (n=15) |
|---|---|---|---|---|---|
| **5-50 m** | | | | | |
| %MEP | 0.35 (0.2-0.5) | 0.53 (0.2-0.8) | 1.23 (0.6-1.8) | 0.87 (0.4-1.3) | 3.20 (2.1-5.6) |
| AI | 0.05 (0.03-0.16) | 0.08 (0.04-0.12) | 0.10 (0.07-0.15) | 0.15 (0.08-0.24) | 0.08 (0.07-0.10) |
| %detritus | 16.08 | 26.46 | 47.54 | 38.87 | 71.47 |
| **50-150 m** | | | | | |
| %MEP | 1.26 (0.8-2.0) | 1.29 (0.9-1.7) | 7.76 (4.7-9.1) | 6.08 (3.7-7.4) | 4.44 (3.4-7.4) |
| AI | 0.05 (0.04-0.11) | 0.08 (0.04-0.11) | 0.10 (0.09-0.11) | 0.10 (0.08-0.14) | 0.08 (0.07-0.10) |
| %detritus | 48.14 | 48.73 | 93.64 | 87.53 | 79.67 |
| **150-300 m** | | | | | |
| %MEP | 1.17 (0.7-1.6) | 1.00 (0.6-1.25) | 5.68 (2.0-7.5) | 2.24 (1.5-3.4) | 2.94 (1.8-4.2) |
| AI | 0.05 (0.03-0.16) | 0.07 (0.03-0.13) | 0.11 (0.07-0.14) | 0.16 (0.09-0.25) | 0.08 (0.06-0.11) |
| %detritus | 46.28 | 42.36 | 85.83 | 62.54 | 69.35 |
| **5-300 m** | | | | | |
| %MEP | 0.88 (0.67-1.14) | 0.96 (0.63-1.25) | 6.10 (3.40-7.33) | 4.87 (3.00-5.97) | 3.81 (2.9-5.7) |
| AI | 0.05 (0.04-0.13) | 0.08 (0.04-0.12) | 0.10 (0.09-0.12) | 0.11 (0.08-0.15) | 0.08 (0.07-0.10) |
| %detritus | 39.16 | 41.33 | 87.62 | 81.98 | 75.84 |
| Abund m$^{-3}$ x10$^4$ | 0.64 (0.61-0.68) | 0.99 (0.65-1.37) | 1.82 (1.46-2.01) | 3.69 (2.71-6.86) | 5.43 (3.30-8.35) |
| BioV mm$^3$ m$^{-3}$ x10$^3$ | 0.33 (0.13-1.24) | 0.37 (0.17-1.29) | 0.80 (0.38-1.20) | 1.47 (0.71-2.25) | 2.58 (1.42-3.56) |
| NBSS | y = -0.85x + 3.71 | y = -0.89x + 3.95 | y = -0.85x + 4.25 | y = -0.93x + 4.58 | y = -0.86x + 4.76 |

**Table 2. LOPC particle derived parameters for ION-Tr, ST8 and SAV, for three depth layers. Abund : Total abundance and BioV : total biovolume.**

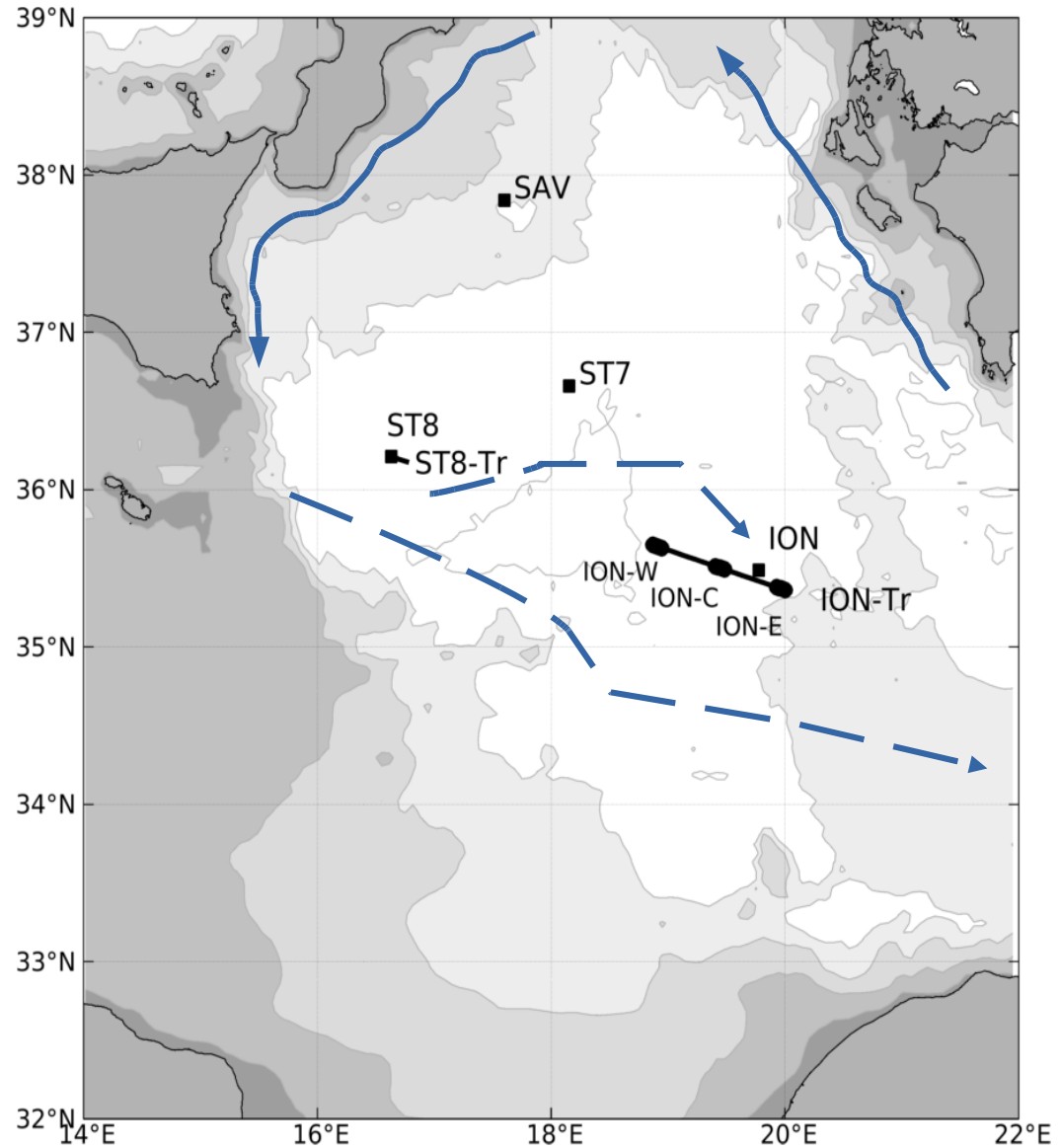

**Figure 1. Bathymetry of the Ionian Sea (isobaths 100, 1000, 2000, 3000, 4000m) with the locations of stations (black squares), transects and transect parts (black lines and dots). Schematic circulation (blue lines with arrows) for cyclonic conditions of the North Ionian Gyre redrawn from Menna et al (2019). The blue dashed line indicates the average AW path.**

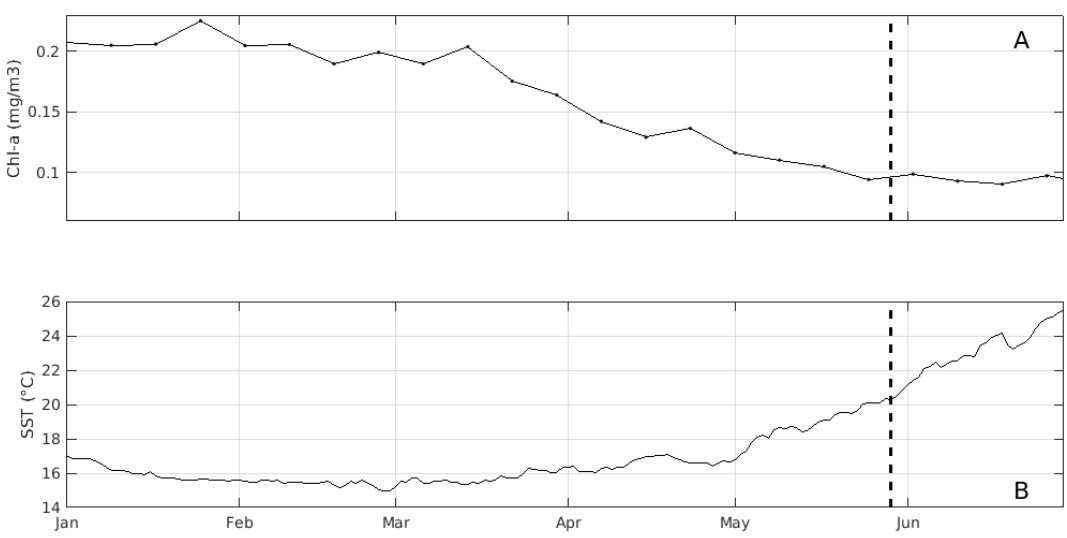

**Figure 2 :  Time series of A) 8-day satellite Chl-a and B) daily SST prior to sampling, averaged over the northern Ionian Sea (16° to 22°E, 34° to 40°N). Sampling at ION-Tr is marked by the dashed vertical line.**

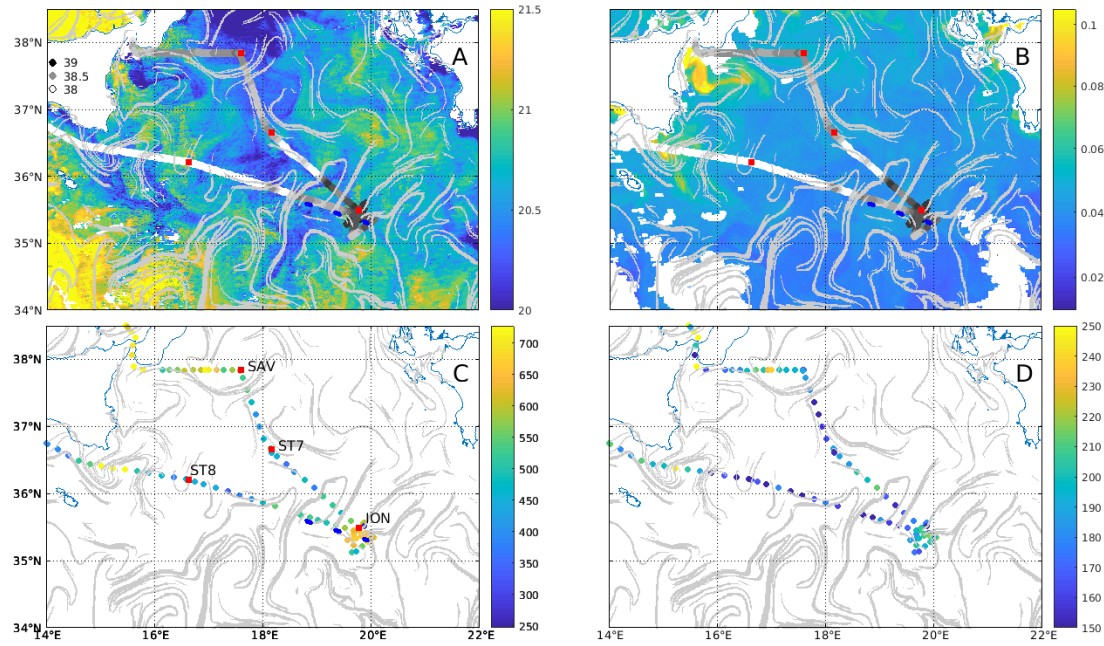

**Figure 3. A)** Satellite SST (May 29-30[th], night, °C). **B)** Chl-a (May 30[th], mg/m³). On panels A and B, underway surface salinity from the TSG is overlaid as dots, darker for saltier water. **C)** Underway surface abundance (cells.cm[-3]) of nanoeukaryotes1 group from cytometry. **D)** Underway surface abundance (cells.cm[-3]) of coccolithophore like group from cytometry. On all panels, FSLE ridges are overimposed in light grey (>0,15 day[-1]). Stations positions are marked as red squares and ION-Tr W, C and E as blue dots as figure 1.

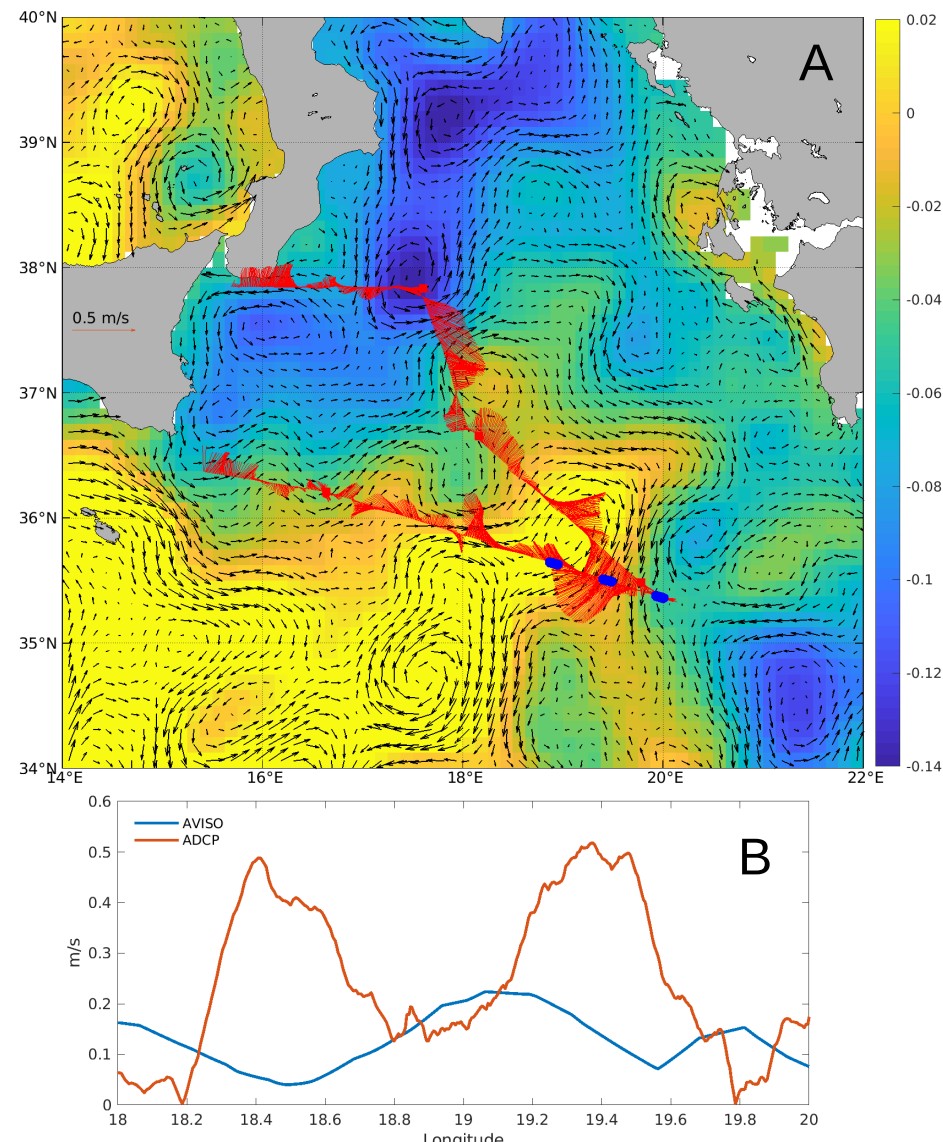

**Figure 4. A) Absolute Dynamic Topography (ADT, color) and surface geostrophic circulation from altimetry averaged over period May 22-30th (black arrows). Underway ADCP currents are overlaid (red arrows). ION-Tr W, C and E marked as blue dots as figure 1. B) Velocity along ION-Tr transect derived from altimetry for May 29th and measured with ADCP .**

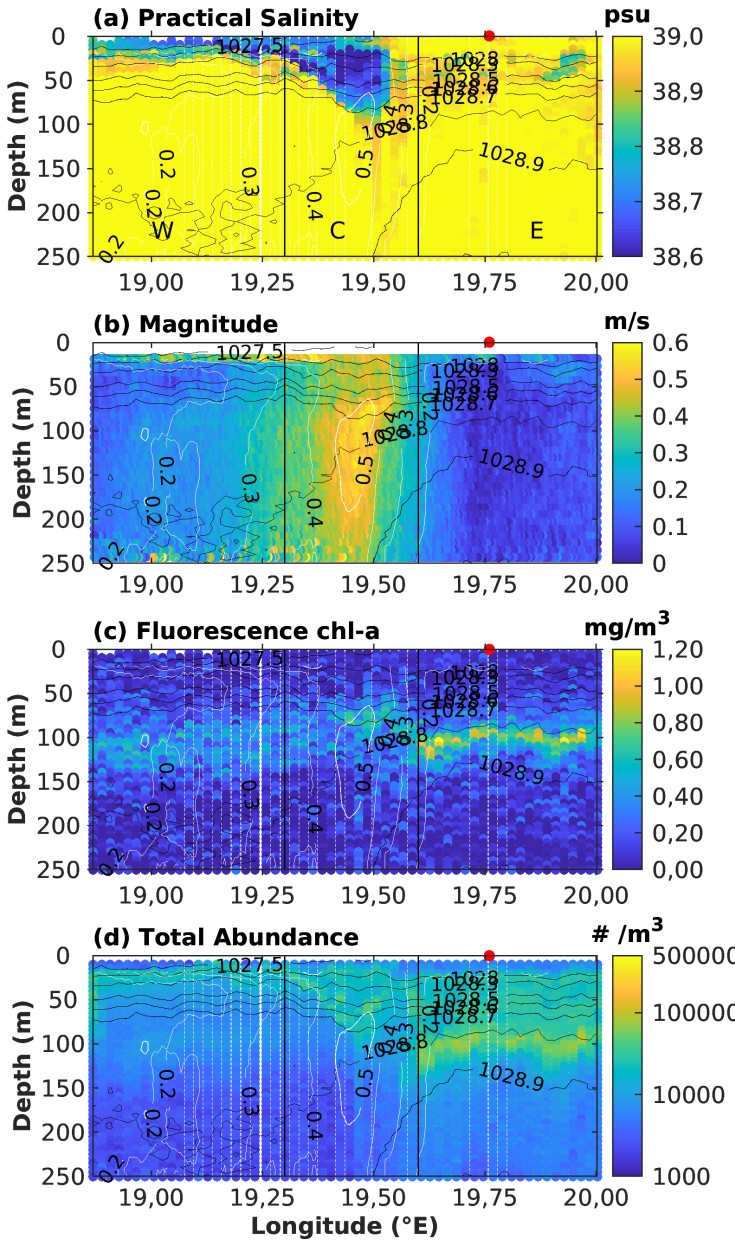

**Figure 5. Variables sampled along ION-Tr transect from MVP, plotted as depth vs longitude. a) Salinity, b) ADCP current speed, c) Fluorescence and d) Total particle abundance. Density and current speed are overlaid as respectively black and white contours on all panels. The three parts of the transect from left to right where respectively ION-W, ION-C, ION-E pseudo-stations are located are delimited by black vertical lines. ION station position is marked as red dot on top of each panel.**

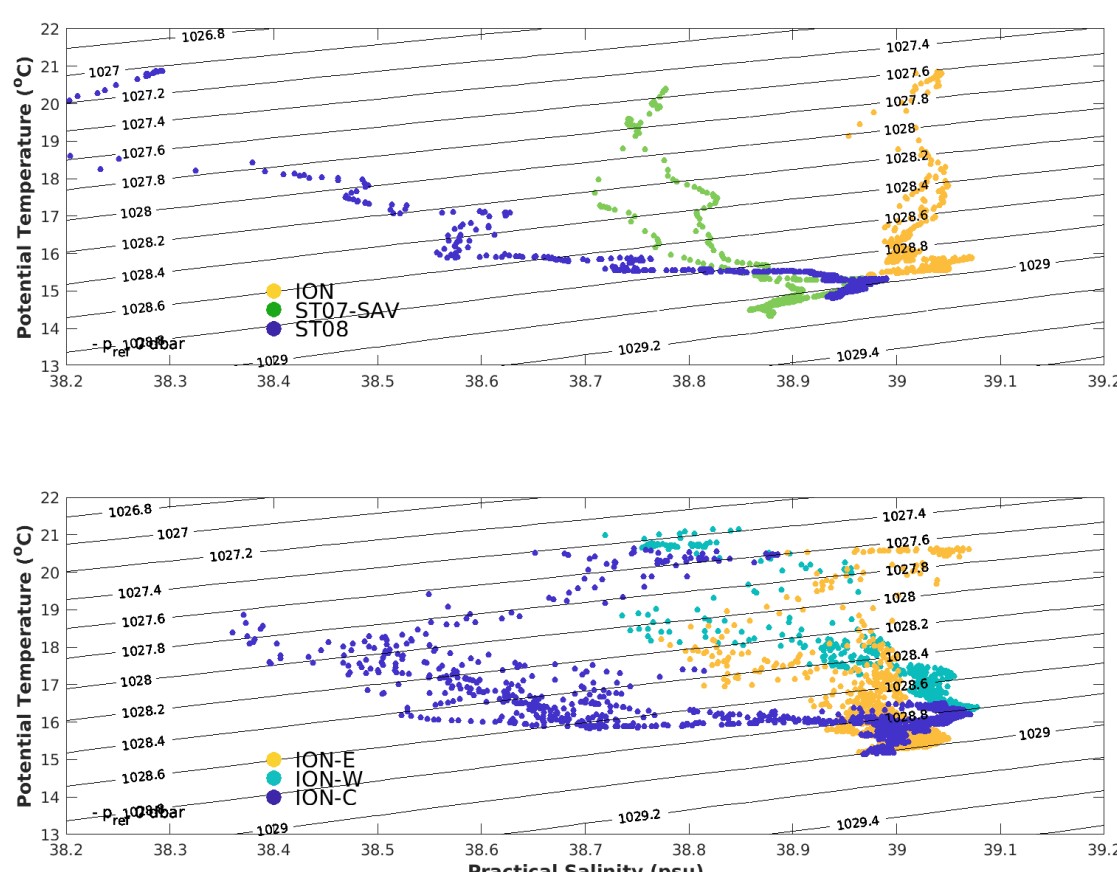

**Figure 6. TS-diagram for stations ION, SAV, ST7, ST8 restricted to 0-300m (top) and for ION-Tr parts (bottom), ION-W, ION-C, ION-E.**

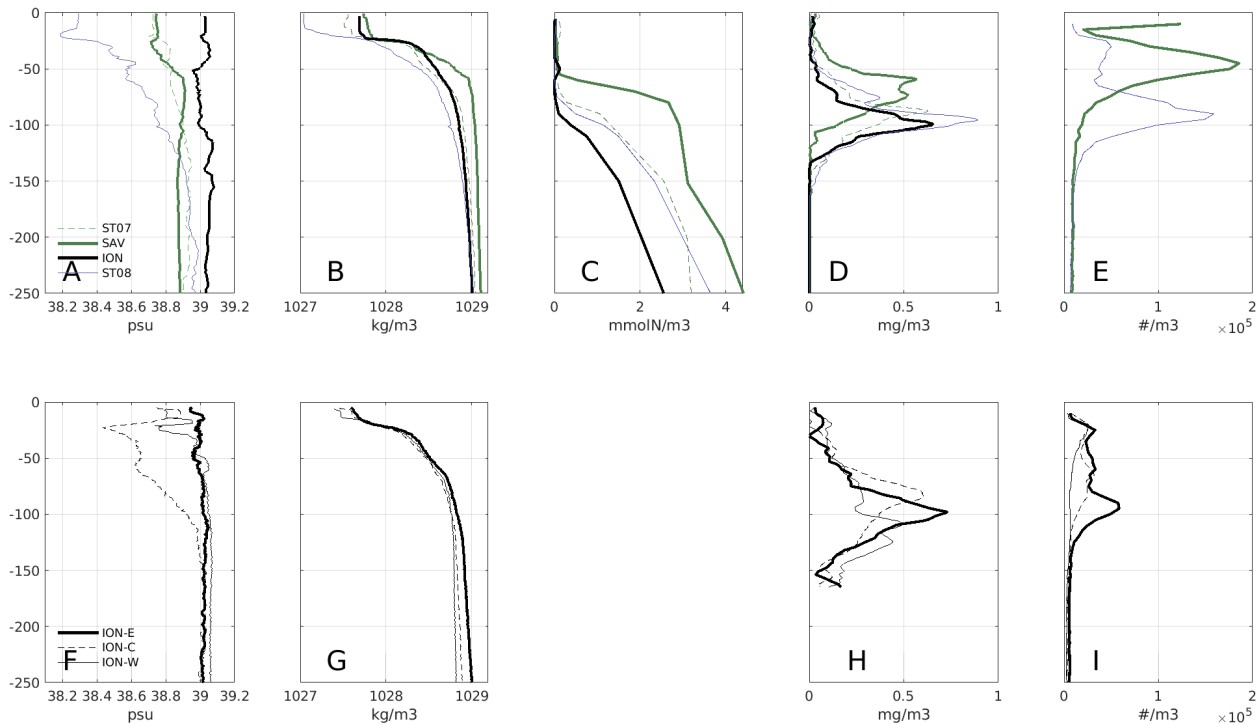

**Figure 7. Vertical profiles of variables at ION, SAV, ST7, ST8 (top) and for ION-W, ION-C, ION-E (bottom). Panels A and F Salinity, B and G density, C Nitrate, D and H Chl-a, E and I total particle abundance. Note that particles were not quantified at ST7. For Chl-a and particles, an average over 3 profiles was computed.**

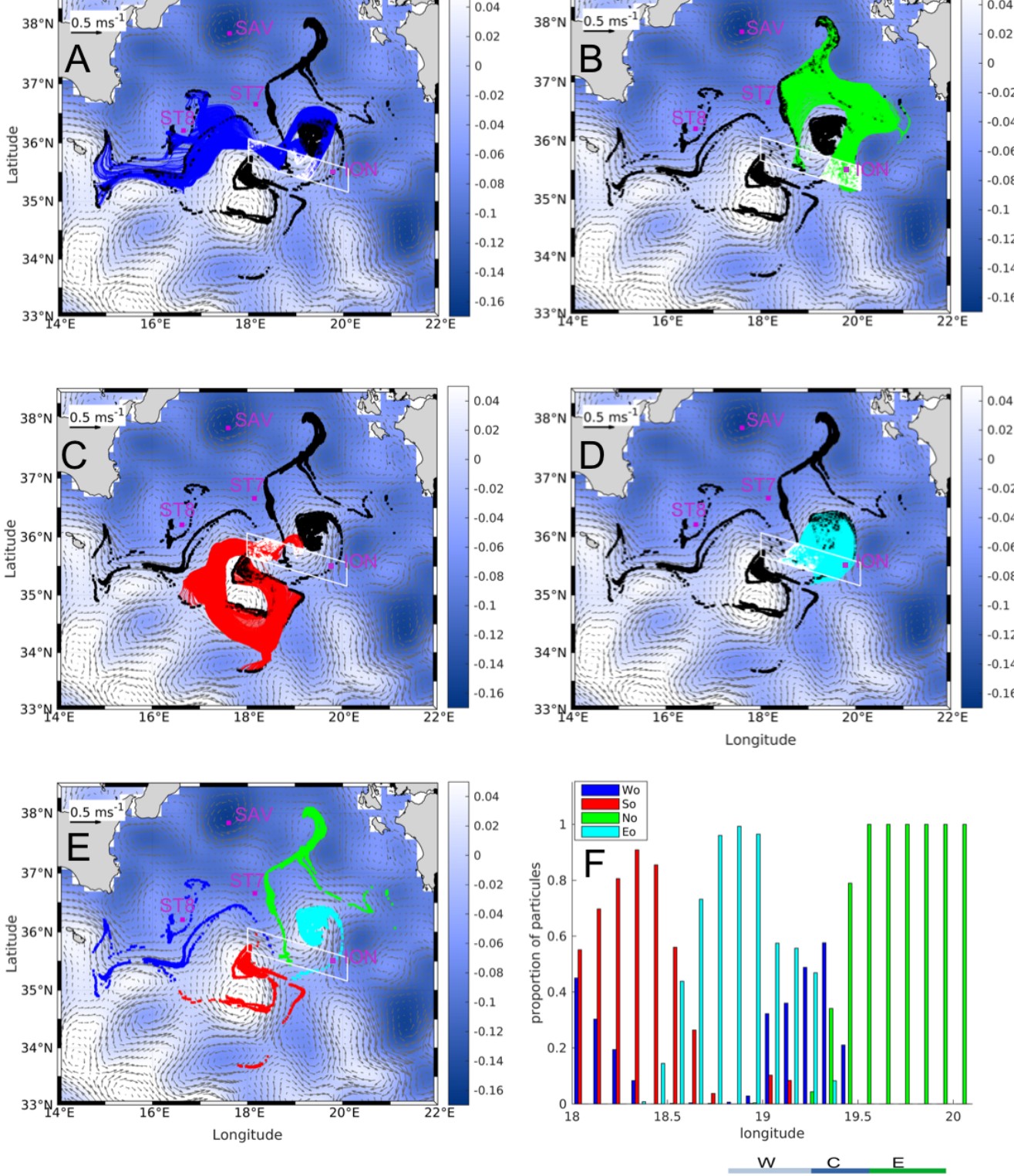

**Figure 8.** Backward Lagrangian trajectories for particles launched along ION-Tr and tracked for one month, overlaid on average ADT (colors) and surface currents (black vectors) averaged for period May 1-30[th]. The particle launch area is indicated by the white polygon. Panels A, B, C, D are trajectories separated according to the four major origins. panel A for Wo, B for No, C for So, D for Eo. Black dots are the particle positions at the end of the backward advection, ie one month before our sampling. E) Final positions of particles for each origin. The pink dots are the stations positions. F) Proportion of each origin along ION-Tr computed for each 0.1° of longitude. The three zones of ION-Tr shown on figure 5 are indicated at the bottom.

**Supplementary material**

https://people.mio.osupytheas.fr/~berline.l/movie_ADT.gif

**Appendix A.** Absolute Dynamic Topography evolution for period Jan 1[st]-June 15[th] 2017. Ship track is marked as thin black line and ION-Tr as thick black line. The anticyclonic eddy sampled during ION-Tr is visible during the whole period.

