# Peer review of "Long distance particle transport to the central Ionian Sea"

_Biogeosciences, 2020_

## Author Comment (AC1)

*We want to thank the reviewer 1 for its detailed reading of our ms and useful suggestions.*

*Our replies to comments are in blue. « Blue quoted text »  is the updated text in the ms.*

*Note that an error was spotted in table 2. Some values for layer 5-300 m have changed, and some sentences in results and discussion have also been changed accordingly.*

**General comments**

The paper examines the heterogeneity observed in the central Ionian Sea with respect to hydrology, productivity and particle properties, using in-situ data collected in 2017 by ship survey as well as remote sensing and Lagrangian modelling data. The sampled biogeochemical and physical data are used to track the various sources and processes that contribute to the marine status of the central Ionian. The manuscript is interesting, however it would benefit by presenting the scope of the research in a more focused way. It is not clear to me if the scope is to show the complexity of the marine environment in the surveyed area or to show different water masses properties and the responsible mechanisms or maybe both.

*We have now clarified the main topic of the ms, which is to show the complexity of the particle distribution in the Ionian Sea, overlooked by the coarse sampling of previous studies.*

General comments per Section are listed here below. The use of English is rather good; some "polishing" could be of help. Finally, the ms needs careful editing since there are many minor errors (see Specific comments).

*We have addressed all these specific comments.*

1 Introduction

Since the manuscript aims at focusing on the particle abundance of different water masses in the Ionian Sea, I would like to see a detailed introduction on the importance of studying particle abundance and their dynamics in the world ocean, the Mediterranean Sea, and the Eastern Mediterranean – Ionian Sea in particular. Following that, I believe a listing of methods employed so far to study particle properties should be presented, and then, the authors can present their own methods, combination of methods applied to better describe the particle properties within different water masses.

*We have reorganized the introduction following these lines, also taking into account remarks from R2.*

[revised manuscript text omitted]

2 Material and Methods

With respect to the methodological design (Section 2), the authors employed several classic and more advanced instruments/methods to study water mass characteristics and particle properties, resulting in the production of a hefty dataset, rather unique, and certainly not previously available for the Ionian Sea marine area.

*We thank the reviewer for recognizing the importance of our new dataset.*

3 Results

Results are adequately presented in brief manner (Section 3) and accompanied by a number of well-designed and informative figures. Given that several parameters and measurements are discussed, whilst certain parameters, such as MEP proportion and AI attenuance index are not so common, a moderately lengthier presentation would be beneficial for the reader.

*We added a more detailed description of MEP and AI in the methods.*

*"LOPC also provides information about the MEPs allowing to compute an attenuance index (AI), which is essentially a normalized opacity index (an AI of 1 means a fully opaque object) although the MEP shape also has an impact (Espinasse et al 2018). The living fraction of particles can be estimated, using a calibration based on comparison with plankton net tows (Espinasse et al., 2018), as this fraction is linked with the percent of MEP with respect to the total particle count (%MEP)."*

With respect to Section 3.2, I do not think that the full depth T-S diagrams (fig. 6) shown are of any significant use in the ms since observations are focusing on the upper water layers. Consider removing. I would only keep the depth-limited vertical profiles (fig. 7).

*As suggested we have removed the deep layers in figure 7.*

The *origin of particles* presented in section 3.4 is not clear. Particles may be transported as part of water masses circulation, but particles are also introduced locally in the water column via atmospheric precipitation and primary production. Since the scope of the paper is about particles, it would be interesting to differentiate between transported and in situ particle origin.

*Thank you for your remark. We now clarified in section 3.4 that we only focus on the 'transported' particle pool in our backtracking simulations. As all particles are transported by circulation, we rather distinguish 'recent' particles and 'old' particles, that have been transported away from their source area. This distinction is discussed in section 4.3 : Considering the late spring season, nutrient depletion in the surface layer and the surface phytoplankton community, we hypothesize that, above the DCM, most particles are 'old', therefore imported through transport.*

*Beyond this hypothesis, distinguishing 'recent' and 'old' particles solely based on the particle properties estimated with the LOPC data is not possible, as the size structure results from various integrated mechanisms that we cannot sort out. Direct observations of particles (bottle, Marine Snow Catcher) may help, but were not available for this study. This point is discussed in section 4.3 Biological history of water masses sampled along ION-Tr.*

A 'particle properties signature' on water masses is a challenge, and I would appreciate if the authors could look deeper into their rich dataset to provide such information, if possible.

*This is an interesting point. Looking at table 2 and fig 7, we can say that*

*i) At the Ionian basin scale, comparing the three water masses seen at ST7-SAV, ST8 and ION, we do have three distinct particle signatures in terms of abundance, vertical profile, %MEP and AI, especially when we restrict the analysis to the top 5-50m layer, which has the most contrasted particle loads.*

*ii) Along ION-Transect, comparing ION-W, ION-C and ION-E we only have two clearly distinct water masses (ION-C and the others). In this case, particle signatures allow to distinguish ION-E and ION-W, with distinct vertical profiles, %MEP and AI. Thus particle properties are complementary to water masses properties as they bring information on their biological history along their pathways.*

*Conversely, distinct ranges of particle properties can identify one single water mass as for ION-E and ION-W. The main difficulty with particle parameters is to define the spatial scale of the ensemble to average (depth layers and profiles), as these parameters have a large variance vertically and horizontally. This is beyond the scope of this study.*

*This discussion was added to the conclusion.*

In several cases across the document, variable units are following the numerical values without a space (e.g. L171: 70m; L180: 20-90m, etc.). Please check and correct throughout the document. In addition, decimal separator used is sometimes a comma (e.g. L136: 0,09 mg.m$^{-3}$) or point (e.g. L149: 38.2). Please decide and correct throughout the document. In the same context, units are sometimes given as fractions (e.g. L175 0.3 m/s) or as exponents (e.g. L177: 5 10$^4$ #.m$^{-3}$). Please correct throughout the document.

*We have checked and homogenized units and numbers throughout the text.*

4 Discussion

The circulation patterns and water mass distribution patterns are presented in the discussion sections 4.1 and 4.2. However, a lot of information seems to fit more nicely at the Results section, and hence should be moved there. That leaves very little a pure discussion to the section. The latter applies also to Section 4.3 on the biological history of water masses.

*We believe that the summary of the different figures is best located in the discussion.  Therefore we moved some introductory sentences to results but left the summarizing parts in the discussion.*

Here, the authors claim that abundant aggregates of diameter larger than 100 μm maybe associated with intense primary production, which, however, does not seem to be the case in the Ionian Sea during the sampling period or some weeks earlier. The authors could consider the presence of Transparent Exopolymer Particles (TEP) as an alternative factor facilitating particle coagulation and thus aggregates formation.

*Thank you for this remark. We have now added a paragraph explaining that the presence of the large aggregates we observed could also result from enhanced concentrations of TEP in the surface layers. (Unfortunately TEP data are not available at the time of this review. They may be available and published in the future.)*

*"LOPC detects particles of ESD >100µm (median ESD was 130-145 µm along ION-Tr). Typically, such large aggregates form as a result of intense primary production (Briggs et al., 2020; Martin et al., 2011). In addition, the presence of transparent exopolymer particles (so called TEP) represents an important upper ocean process by which dissolved organic matter is converted into particles. Besides, these compounds act as glue enhancing small particles aggregation and leading to the formation of marine snow aggregates (Engel et al 2004) of the size we observe here."*

5 Conclusions

This Section briefly summarizes all findings: heterogeneity, water masses origin, particle behavior and implications. What about the initial hypothesis, can we characterize and identify water masses by their particle properties and abundance?

*We have added our point on the added value (section 3.4) of particle properties.*

**Specific comments:**

**Abstract:** The abstract needs some restructuring. I suggest the last sentence (lines 27-30) could move at the beginning of the abstract. I leave this to authors' decision.

*We followed this suggestion. The abstract was updated in accordance to the changes made to the text:*

*"Particle abundance in situ measurements are useful to discriminate water masses and derive circulation, together with T-S properties. In the upper layers of the Ionian Sea, the fresher Atlantic Waters (AW) recently crossing the Sicily channel meet the resident and saltier AW which have cyclonically circulated in the Eastern basin and come back modified after evaporation and eventually cooling. In May 2017, during the PEACETIME cruise, fluorescence and particle content sampled at high resolution revealed unexpected heterogeneity in the central Ionian. Surface salinity measurements, together with altimetry-derived and hull-mounted ADCP currents, describe a zonal pathway of AW entering the Ionian Sea, consistent with the so-called cyclonic mode in the North Ionian Gyre. The ION-Tr transect, located ~19-20°E- ~36°N turned out to be at the crossroad of three water masses, mostly coming from the west, north and from an isolated anticyclonic eddy northeast of ION-Tr. Using Lagrangian numerical simulations, we suggest that the contrast in particle loads along ION-Tr originates from particles transported from these three different water masses. Waters from the west, identified as AW carried by a strong southwestward jet, were moderate in particle load, probably originating from the Sicily channel. Water mass originating from the north was carrying abundant particles, probably originating from the northern Ionian, or further from the south Adriatic. Waters from the eddy, depleted in particles and Chl-a may originate from south of Peloponnese, where the Pelops eddy forms. The central Ionian Sea hence appears as a mosaic area, where waters of contrasted biological history meet. This contrast is particularly clear in spring, when blooming and non-blooming areas co-occur. This work showed that interpreting the complex dynamics of physical-biogeochemical coupling from discrete measurements made at isolated stations at sea is a challenge, that can be overcome by the combination of multiparametric in situ measurements at high resolution with remote sensing and Lagrangian modeling."*

Line 12 and throughout the text. MAW and AW acronyms are both used for Atlantic water in the ms. Since (mostly in older literature) MAW stands for Modified Atlantic Water and in the ms it

stands for Mediterranean Atlantic Water (the latter is not used in literature), I strongly suggest that whenever you refer to Atlantic Water you should use the acronym AW.

*We now use AW everywhere as suggested, and removed MAW, as this study is fully in the Mediterranean Sea.*

Lines 12 and 19: What is "young AW"? Is there an "old AW"?

*We changed the sentence to be more explicit : recent AW flowing from the Sicily channel and older AW (modified after its cyclonic circulation in the Eastern Mediterranean).*

Line 20: Try replacing the word "intermediate" with another one to avoid confusion with layer-depth meaning. I suggest "moderate" instead.

*Done*

Line 42: **The** Ionian Sea (check and correct throughout the manuscript)

*Done*

Line 55-59: Please provide information regarding station SAV, which appears to be isolated both from other stations and the transect area. Based on what properties was SAV selected?

*During the cruise, a drifter was launched at the SAV station. Then a CTD cast was carried out. In our study, the SAV station is important as it represents a northern Ionian situation. We now justify this choice of station in the material and methods.*

Line 61: Change "salinity" to "conductivity" which is what the sensor directly reads. *Done*

Line 62: State if the fluorometer was calibrated to report chlorophyll-a concentration in mg m$^{-3}$ *Done*

Line 63: The primary parameter measured is light transmission (LT, %) and then the beam attenuation coefficient (c, m$^{-1}$) is estimated according to the path-length (L) of the transmissometer according to: c = -1/L ln(LT/100). Please describe in more detail or give references. *Done*

Line 70: WetLabs *Done*

Line 78: I cannot find ST8-Tr in figure 1. *We now have added ST8-Tr to figure 1.*

Line 70: What is the depth reached by LOPC? Give a range. *Done (it was ~300m)*

Line 85: TSG most probably means ThermoSalinoGraph. Please expand the acronym to its proper meaning. *Done*

Line 122: Same as above for SVP. *Done*

Line 123: Consider replacing "numerical" with "simulated". "Numerical particles" sounds a bit strange. *Done*

Line 131: Please provide link for the database. *Done*

Line 136: Provide value for maximum chlorophyll-a concentration *Done*

Line 147: Please provide a range for "low, high" salinity. *Done*

Line 159: I do not think that the altimetry shows anticyclonic circulation "across" the basin.

*We have rephrased the sentence: anticyclonic circulation in the southwestern part of the basin.*

Line 163: Please state that ~0.4° refers to longitude. Same throughout the text. *Done*

Line 168: Maybe 'water mass' properties *Done. changed to 'water masses and biological properties'*

Lines 169-171: This is confusing. You could say that the surface layer which reaches depths up to 30 or 70 m in this or that area, carries a low salinity AW signal, below which salinity increases to more than 39 etc. Please rephrase. *The sentence was rephrased.*

Line 180: term**s** *Done*

Line 186: The density values refer to potential density? Please state in the text. Moreover here and throughout the text density units are missing. Please correct. *Done*

Line 196: Replace "carousel" with "CTD". I do not think that the water sampling at discrete levels from the carouse/rosette can produce continuous profiles as in fig. 7. These are most certainly the product of CTD sensors.

*Done (actually only nitrate was measured from bottle)*

Line 216: Paragraph 3.4 is a bit repetitive. Please restructure.

*The paragraph was rewritten removing redundancy.*

Lines 261-2-3: I am not sure if the presentation of specific density values here has any particular meaning. Consider omitting.

*The particle vertical distribution closely matches the density isolines (see figure 5d), so we kept our presentation as is.*

Line 318: Change "*This eddy may have trapped waters from the eastern Ionian, more oligotrophic (Casotti et al 2003),….*" to "*This eddy may have trapped waters from the more oligotrophic eastern Ionian (Casotti et al 2003),….*".

*Done*

Line 321: The acronym UVP is used here with no explanation. Please add full meaning: Underwater Video Profiler.

*Done*

Line 364:  Correct 4 to 5

*Done*

**Figure captions:**

Fig. 1: What is the difference between full and dotted lines?

*The dotted line is the average AW path. This was added to the legend.*

Fig. 7: Please add variable names at all x axes

*Done*

Fig. 8: In the phrase "*Black dots are the final positions*", the word "final" possibly refers to the positions of the particles calculated at the beginning of one month before the sampling. If that is true, then "final" should be replaced with "starting" or something similar to avoid confusion.

*The legend was changed as follows: « Black dots are the particle positions at the end of the backward advection, ie one month before our sampling. »*

---

## Author Comment (AC2)

*We want to thank the reviewer 2 for its detailed reading of our ms and useful suggestions.*

*Our replies to comments are in blue. « Blue quoted text » is the updated text in the ms.*

*Note that an error was spotted in table 2. Some values for layer 5-300 m have changed, and some sentences in results and discussion have also been changed accordingly.*

**R2**

**General Comments:**

This paper focus on the oceanographic and biological data collected during the PEACETIME cruise in the Ionian Sea in May 2017. The results focus on the heterogeneity of the water masses salinity, fluorescence and particle content observed along the ION-tr transect, suggesting their different origins and connecting them with the surface dynamics of the Ionian Basin. The ION-tr transect results in an interesting crossroads between three different water masses; their presence and characteristics are linked to the basin-scale circulation of the Ionian Sea.

I find this work very interesting and certainly suggest its publication, but I also believe that the authors can still work on improving the description of the context and results. The introduction should be used to explain the importance of this kind of studies and to introduce the dynamics and the surface circulation of the Ionian Sea, in order to give the reader the right elements to better understand and interpret the results. In this form, it becomes very difficult to follow the descriptions without a deep knowledge of the study area. The results of Lagrangian numerical simulation are an important point of this work, and the criteria chosen for the simulations should be explained in more detail.

*We have developed the context in the introduction to provide background information on the Ionian Sea circulation and why studying particle distribution is important:*

[revised manuscript text omitted]

***Regarding the criteria chosen for the simulations, we have detailed our choices in the new version of the Material and Methods section.***

*"The simulated particle positions were initialized in a polygon around the ship's track on May 29[th] (Longitude from 18° to 20.1°, Latitude from 35.4° to 35.9° with a 0.2° widening in Latitude). This polygon surrounds the actual track, to consider the uncertainty associated to the spatio-temporal resolution of the satellite-derived current velocity field. 10000 particles (sufficient for a comprehensive sampling of the polygon area and for saving computing time) randomly seeded inside the launch area were advected backward for one month. This duration was chosen as a trade-off between particle source identification, error accumulation along the trajectories and*

*residence time in the Ionian Sea (39 days according to Celentano et al 2020). The particle trajectories and final positions were analyzed to determine the main source areas. "*

**Specific comments:**

**Introduction:**

I like short, concise introductions, but I find this one too essential. It would be interesting to have a brief discussion of the main results in the literature and some considerations on the importance to study particle distribution.

*See above.*

Lines 47-48: these two sentences seem a repetition.

*Right. We changed this paragraph, including more background.*

Lines 51:53: I suggest a broader explanation of the background (PEACETIME cruise) and of the objectives of this study. This sentence rather than explaining the objectives seems to anticipate the results in a cryptic way.

*We added details for the PEACETIME cruise in the methods. The objectives are better explained (See above).*

**Materials and Methods:**

Figure 1: I suggest reproducing the figure in a more accurate way. The transect ST8-tr is not labelled in the figure; the label of the ION-tr transect is positioned away from the transect itself; use different colours for circulation patterns and transects.

*We updated the figure adding ST8-Tr and arrows to be more explicit.*

**Results:**

Lines 135-138: Figure 2 is introduced at the beginning of the results without context (it is unclear where the time series was derived from and why). I suggest inserting a paragraph at the beginning of the section that introduces and contextualizes the results explained below. For example, the authors could start by describing the Ionian surface circulation during PEACETIME cruise and explaining why that area was chosen the averages of SST and Chl-a. In the caption of Figure 2, the authors talk about an average on the "northern Ionian Sea", but actually the selected area involves almost all the Ionian Sea.

*Following the reviewer's suggestions, we added the following sentence to introduce the figure and the choice of the region. We prefer not to discuss here the surface circulation as it is shown on figure 4. The caption of the figure was also modified accordingly.*

*"Considering the surface circulation over the northern Ionian Sea (figure 1), averages of satellite Chl-a and surface temperature for the region 16° to 22°E, 34° to 40°N were computed for the period before our sampling."*

*We restricted the region of average (34-40°N) to fit the region of influence of our transect.*

Lines 143-144: it should be better explained to the reader how to recognize these cyclonic and anticyclonic structures in Fig. 3a and 3b.

*We now included the positions of the main eddies in the text: (anticyclones located at 34.7°N 17.5°E, 35.7°N 19°E, and cyclones located at 37.5°N 16E, 38°N 17.5°E and 34.5°N 21.3E)*

Line 153: this sentence is not clear for me! "Sufficient abundance" of what?

*We rephrased the sentence :*

*«showing contrasted distribution and abundant enough for the counts to be accurate.»*

*See table 2 in Leroux et al (2017) http://dx.doi.org/10.1016/j.pocean.2017.10.010*

Line 155: add "…of Sicily, where lower salinities are associated to large abundance."

*We changed the sentence as suggested.*

Line 163: add also a comment about the difference in speeds between AVISO and ADCP currents.

*We added : «current magnitude estimated from AVISO was about half the value of ADCP derived current magnitude»*

Section 3.3: It would be easier for the reader to follow this description by adding in the text an indication of the sub-plots referred to.

*We now refer to the subplots in the text.*

Section 3.4: The results of this figure are essential to understanding the discussions and conclusions sections of this work. For this reason I believe that the authors should explain more accurately what we see in the different panels of Figure 8 and then focus on the results of panels E and F. The rectangle enclosing the location of the ION-tr transect in Figure 8E is really hard to see.

*Thank you for this remark. We now explain further the panels in the results.*

*"Figure 8 show the full trajectories of particles originating along the ION transect, splitted in four panels according to the main path followed by particles and therefore origin. Each color identifies a distinct path. Panel E show the particle positions one month before our sampling, colored according to the origin. "*

*We also updated figure 8 to make it clearer and larger. The legend was changed as follows:*
*« Black dots are the particle positions at the end of the backward advection, ie one month before our sampling."*

**Technical Comments:**

Line 85: the acronym TSG is not defined

*ThermoSalinoGraph (now in full in text).*

Line 111: check the spatial resolution of the SST.

*Thank you for this remark. The SST data was at 1km resolution indeed. It is now corrected in the methods.*

---

## Author Response (AR1)

*Please find our answers to the editor remarks in blue.*

Dear authors,
the answer to comments appropriately addressed comments from both reviewers. At this stage I have only a few minor comments:

- Do, among others, the flow cytometer and TEP data (Marañon et al., Zäncker et al., in the same issue) corroborate the conclusions here?

We have looked at the papers of the Special Issue and cited the few ones linked to our study. Observations of Marañón et al (2020) of high bacterial production in the layer 0-100 m were in line with abundant particles in this layer, and stable nanoeukaryotes abundances (in discussion section 4.3.3). Barbieux et al (2021) observations were also in line with ours.

Unfortunately Zäncker et al reports TEP data only from the sea surface microlayer, which is not representative of the water column, thus we cannot compare to our observations.

- It is somewhat difficult to follow the description of circulation in the text (Discussion section 4.1). Marking salient relevant features in Fig. 4 (such as the NIG vs. eddies) would be helpful.
As figure 4 is already complex, I prefer not to overload it with additional arrows and text. The schematic circulation is shown on figure 1, and legend now mentions NIG.

Fig. 1: replace "dark blue arrows" with dotted blue lines with arrows".
Fig 1 legend was updated accordingly

Fig. 5: it is difficult to distinguish isolines showing density from those representing current "magnitude" in the panels. Maybe use different colors or report these in different panels, respectively. Please define "current magnitude" (transport in Sv or speed?) and give units for both parameters in the legend.
Fig 5 was updated with white contours for ADPC currents. We now use "Current speed" in the legend (in m/s).

Fig.8: Is it possible to add the location (polygon) where particles have been injected in the panels? What are the pink squares in the figure (I imagine station positions?).
Fig 8 was fully updated and the seeding locations are now clearly indicated as white polygon. Pink squares are described in the legend.

---

## Author Response (AR2)

**bg-2020-481**

**Response to edits and comments from editor**

Thank you for the detailed reading.

We have applied all edits, except L262: kept 'originating' as it is really what is meant.

Regarding comments

L39 Done

L86 Done

L114 Information was already in the text

L149 Done

L156. Reworded: latitude from 35.4° to 35.9° N with an extension of 0.2° north and south of the track.

L202 Removed.

L233 These are single profiles for salinity and density and an average over 3 profiles for Chl-a and particles. Now added in the legend.

L291: Both SAV and ST8 are located in the central NIG. Now explained in the text.

Figure 3. Done

Figure 4. Unchanged, as station names would mask vectors

Figure 6. Done

Figure 8. Done